# ZenDet: Revisiting Efficient Object Detection Backbones from Zero-Shot Neural Architecture Search

## Abstract

In object detection models, the detection backbone consumes more than half of the overall inference cost. Recent researches attempt to reduce this cost by optimizing the backbone architecture with the help of Neural Architecture Search (NAS). However, existing NAS methods for object detection require hundreds to thousands of GPU hours of searching, making them impractical in fast-paced research and development. In this work, we propose a novel zero-shot NAS method to address this issue. The proposed method, named ZenDet, automatically designs efficient detection backbones without training network parameters, reducing the architecture design cost to nearly zero yet delivering the state-of-the-art (SOTA) performance. Under the hood, ZenDet maximizes the differential entropy of detection backbones, leading to a better feature extractor for object detection under the same computational budgets. After merely one GPU day of fully automatic design, ZenDet innovates SOTA detection backbones on multiple detection benchmark datasets with little human intervention. Comparing to ResNet-50 backbone, ZenDet is $+2.0\%$ better in mAP when using the same amount of FLOPs/parameters and is $1.54$ times faster on NVIDIA V100 at the same mAP. Code and pre-trained models will be released after publication.

## 1 Introduction

Seeking better and faster deep models for object detection is never an outdated task in computer vision. The performance of a deep object detection network heavily depends on the feature extraction backbone (Li et al., 2018; Chen et al., 2019b). Currently, most state-of-the-art (SOTA) detection backbones (He et al., 2016; Xie et al., 2017; Zhu et al., 2019) are designed manually by human experts which could take years to develop. Since the detection backbone consumes more than half of the total inference cost in many detection frameworks, it is critical to optimize the backbone architecture for better speed-accuracy trade-off on different hardware platforms varying from server-side GPUs to mobile-side chipsets. To reduce the manual design, Neural Architecture Search (NAS) has emerged to facilitate the architecture design. Various NAS methods have demonstrated their efficacy in designing SOTA image classification models (Zoph et al., 2018; Liu et al., 2018; Cai et al., 2019; Tan & Le, 2019). These successful stories inspire recent researchers to use NAS to design detection backbones (Chen et al., 2019b; Du et al., 2020; Jiang et al., 2020) in an end-to-end way.

The existing NAS methods for detection backbone design are all training-based, meaning that they need to train network parameters to evaluate the performance of candidates on the target dataset, taking a long time and consuming huge hardware resources. Hardware consumption and long time searching make these training-based methods inefficient in modern fast-paced research and development. To reduce the training cost, training-free methods are proposed recently, also known as *zero-shot NAS* in previous literatures (Tanaka et al., 2020; Mellor et al., 2021; Chen et al., 2021b; Lin et al., 2021). The zero-shot NAS predicts network performance without training network parameters therefore is way faster than training-based NAS. As a relatively new technique, existing zero-shot NAS methods are mostly validated on image classification datasets. Applying zero-shot NAS to object detection backbone design is still an intact challenge.

In this work, we present the first effort of introducing zero-shot NAS technique to design efficient object detection backbones. We show that directly transferring existing zero-shot NAS methods from image classification to detection backbone design will encounter fundamental difficulties. While image classification network only needs to predict the class probability, object detection network needs to additionally predict the bounding boxes of multiple objects, making the direct architecture transfer sub-optimal. To this end, a novel zero-shot NAS method, termed ZenDet, is proposed for searching object detection backbones. The key idea behind ZenDet is inspired by the *Principle of Maximum Entropy* (PME) (Reza, 1994; Kullback, 1997; Brillouin, 2013). Informally speaking, when a network is formulated as an information processing system, its capacity is maximized when its differential entropy (Shannon, 1948) achieves maximum under budget constraints, leading to a better feature extractor for object detection. Based on this observation, ZenDet maximizes the differential entropy of detection backbones by searching for the optimal configuration of network depth and width without training network parameters.

The above strategy raises two technical challenges. **The first challenge is how to estimate the differential entropy of a deep network**. The exact computation of differential entropy requires knowing the precise probability distribution of deep features in high dimensional space which is difficult to estimate in practice. To address this issue, ZenDet estimates the Gaussian upper bound of the differential entropy which only requires computing the variance of the feature maps. **The second challenge is how to efficiently capture objects of different sizes**. In object detection benchmark datasets such as MS COCO Lin et al. (2014), the distribution of object size is data-dependent and non-uniform. To bring in this prior knowledge in backbone design, we introduce the *Multi-Scale Entropy Prior* (MSEP) in backbone entropy estimation to capture different-scale objects. We find that the MSEP improves the detection performance significantly. The overall computation of ZenDet only requires one forward inference of the detection backbone at initialization therefore is nearly zero-cost comparing to previous backbone NAS methods.

The main contributions of this work are summarized as follows:

- Based on the entropy theory, the multi-scale entropy prior is present to rank the expressivity of the backbone instead of training on the target datasets, speeding up searching.
- While using less than one GPU day and 2GB GPU memory, ZenDet achieves competitive performance over other NAS methods on COCO with at least 50x times faster.
- ZenDet is the first zero-shot NAS method designed for object detection with SOTA performance in multiple benchmark datasets under multiple popular detection frameworks.

## 2 RELATED WORK

**Backbone for Object Detection** Recently, object detectors composed of backbone, neck and head have become increasingly popular due to their effectiveness and high performance (Lin et al., 2017a;b; Tian et al., 2019; Li et al., 2020; 2021). Prevailing detectors directly use the backbone designed for image classification to extract multi-scale features from an image, such as ResNet (He et al., 2016), ResNeXt (Xie et al., 2017) and Deformable Convolutional Network (DCN) (Zhu et al., 2019). Nevertheless, the backbone migrated from image classification may be suboptimal in object detection (Ghiasi et al., 2019). To tackle the gap, many architectures are end-to-end designed for object detection, including Stacked Hourglass Newell et al. (2016), FishNet Sun et al. (2018), DetNet Li et al. (2018), HRNet Wang et al. (2020a) and so on. Albeit with improved performance, these hand-crafted detection architectures heavily rely on human labor and tedious trial-and-error processes.

**Neural Architecture Search** Neural Architecture Search (NAS) is initially developed to automatically design network architectures for image classification models (Zoph et al., 2018; Liu et al., 2018; Real et al., 2019; Cai et al., 2019; Lin et al., 2020; Tan & Le, 2019; Lin et al., 2021). Using NAS to design object detection models has not been well studied. Currently, existing detection NAS methods are all training-based methods. Some methods focus on searching detection backbones, such as DetNAS (Chen et al., 2019b), SpineNet (Du et al., 2020) and SP-NAS (Jiang et al., 2020), while the others focus on searching FPN neck, such as NAS-FPN (Ghiasi et al., 2019), NAS-FCOS (Wang et al., 2020b) and OPANet (Liang et al., 2021). These methods require training and evaluation on the target datasets which is intensive in computation. ZenDet distinguishes itself as being the first zero-shot NAS method for the backbone design of object detection.

## 3 PRELIMINARY

In this section, we first formulate a deep network as a system with continuous state space. Then we define the differential entropy of this system and show how to estimate this entropy via its Gaussian upper bound. Finally we introduce the basic concept of vanilla network search space for designing our detection backbones.

**Continuous State Space of Deep Networks**     A deep network $F(\cdot) : \mathbb{R}^d \to \mathbb{R}$ maps an input image $x \in \mathbb{R}^d$ to its label $y \in \mathbb{R}$. The topology of a network can be abstracted as a graph $\mathcal{G} = (\mathcal{V}, \mathcal{E})$ where the vertex set $\mathcal{V}$ consists of neurons and the edge set $\mathcal{E}$ consists of spikes between neurons. For any $v \in \mathcal{V}$ and $e \in \mathcal{E}$, $h(v) \in \mathbb{R}$ and $h(e) \in \mathbb{R}$ present the values endowed with each vertex $v$ and each edge $e$ respectively. The set $\mathcal{S} = \{h(v), h(e) : \forall v \in \mathcal{V}, e \in \mathcal{E}\}$ defines the continuous state space of the network $F$.

According to the Principle of Maximum Entropy, we want to maximize the differential entropy of network $F$, under some given computational budgets. The entropy $H(\mathcal{S})$ of set $\mathcal{S}$ measures the total information contained in the system (network) $F$, including the information contained in the latent features $H(\mathcal{S}_v) = \{h(v) : v \in \mathcal{V}\}$ and in the network parameters $H(\mathcal{S}_e) = \{h(e) : e \in \mathcal{E}\}$. As for object detection backbone design, we only care about the entropy of latent features $H(\mathcal{S}_v)$ rather than the entropy of network parameters $H(\mathcal{S}_e)$. Informally speaking, $H(\mathcal{S}_v)$ measures the feature representation power of $F$ while $H(\mathcal{S}_e)$ measures the model complexity of $F$. Therefore, in the remainder of this work, the differential entropy of $F$ refers to the entropy $H(\mathcal{S}_v)$ by default.

**Entropy of Gaussian Distribution**     The differential entropy of Gaussian distribution can be found in many textbooks such as (Norwich, 1993). Suppose $x$ is sampled from Gaussian distribution $\mathcal{N}(\mu, \sigma^2)$. Then the differential entropy of $x$ is given by

$$H^*(x) = \frac{1}{2}\log(2\pi) + \frac{1}{2} + H(x) \qquad H(x) := \log(\sigma) \,. \tag{1}$$

From Eq. 1, the entropy of Gaussian distribution only depends on the variance. In the following, we will use $H(x)$ instead of $H^*(x)$ as constants do not matter in our discussion.

**Gaussian Entropy Upper Bound** Since the probability distribution $\mathbb{P}(\mathcal{S}_v)$ is a high dimensional function, it is difficult to compute the precise value of its entropy directly. Instead, we propose to estimate the upper bound of the entropy, given by the following well-known theorem (Cover & Thomas, 2012):

**Theorem 1.** *For any continuous distribution $\mathbb{P}(x)$ of mean $\mu$ and variance $\sigma^2$, its differential entropy is maximized when $\mathbb{P}(x)$ is a Gaussian distribution $\mathcal{N}(\mu, \sigma^2)$.*

Theorem 1 says that the differential entropy of a distribution is upper bounded by a Gaussian distribution with the same mean and variance. Combining this with Eq. (1), we can easily estimate the network entropy $H(\mathcal{S}_v)$ by simply computing the feature map variance and then use Eq. (1) to get the Gaussian entropy upper bound for the network.

**Vanilla Network Search Space** Following previous works, we design our backbones in the vanilla convolutional network space (Li et al., 2018; Chen et al., 2019b; Du et al., 2020; Lin et al., 2021) because this space is widely adopted in detection backbones and is used as a prototype in theoretical literature (Poole et al., 2016; Serra et al., 2018; Hanin & Rolnick, 2019). A vanilla network is stacked by multiple convolutional layers followed by RELU activations. Auxiliary components such as residual link (He et al., 2016), Batch Normalization (BN) (Ioffe & Szegedy, 2015) and Squeeze-and-Excitation (SE) (Hu et al., 2018) are all removed **during the search and only during the search (More details in Appendix E).** These removed auxiliary components are plugged in back after the search. Therefore, the final architecture for training still has these components.

Consider a vanilla convolutional network with $D$ layers of weights $\mathbf{W}^1, ..., \mathbf{W}^D$ whose output feature maps are $x^1, ..., x^D$. The input image is $x^0$. Let $\phi(\cdot)$ denote the RELU activation function. Then the forward inference is given by

$$x^l = \phi(h^l) \quad h^l = \mathbf{W}^l * x^{l-1} \quad for \; l = 1, ..., D \,. \tag{2}$$

For sake of simplicity, we set the bias of the convolutional layer to zero.

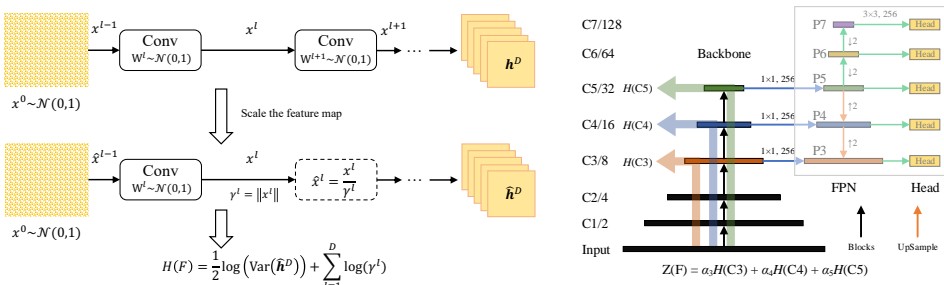

(a) Single-scale entropy score with rescaling    (b) Multi-scale entropy score for detection

Figure 1: Computational graph of entropy score for detection backbone with multi-scale features.

## 4 ZENDET-NAS FOR OBJECT DETECTION

In this section, we first describe how to compute the differential entropy of the single-scale feature for very deep vanilla networks with the re-scaling trick. Then we introduce the Multi-Scale Entropy Prior (MSEP) to better capture the prior distribution of object size in ZenDet. Finally, we present the complete ZenDet-NAS under the Evolutionary Algorithm (EA) framework.

### 4.1 SINGLE-SCALE ENTROPY FOR VERY DEEP BACKBONES

To compute the entropy of the detection backbone, we initialize all backbone parameters by the standard Gaussian distribution $\mathcal{N}(0, 1)$. Then we randomly generate an image filled with the standard Gaussian noise and perform forward inference. Based on the discussion in Section 3, the (Gaussian upper bound) entropy $H(F)$ of the network $F$ is then given by

$$H(F) = \frac{1}{2} \log(\text{Var}(\boldsymbol{h}^D)) . \tag{3}$$

Please note that the variance is computed on the last pre-activation feature map $\boldsymbol{h}^D$.

For very deep networks, directly using Eq. (3) might cause a numerical overflow. The same problem is also observed in Zen-NAS (Lin et al., 2021). Inspired by the analysis in Zen-NAS where the authors show that the gradient norm is scale-invariant to BN layers after variance compensation, we propose a more straightforward solution. We directly re-scale each feature map $\boldsymbol{x}^l$ by some constants $\gamma^l$ during inference, that is $\boldsymbol{x}^l = \phi(\boldsymbol{h}^l)/\gamma^l$, and then compensate the entropy of the network by

$$H(F) = \frac{1}{2} \log(\text{Var}(\hat{\boldsymbol{h}}^D)) + \sum_{l=1}^{D} \log(\gamma^l) . \tag{4}$$

The values of $\gamma^l$ can be arbitrarily given as long as the forward inference does not overflow or underflow. In practice, we find that simply setting $\gamma^l$ to the Euclidean norm of the feature map works well. The overall process is illustrated in Figure 1 (a).

**Comparison with Zen-NAS**    Although the definition of vanilla network and solution to overflow are similar, the principles of single-scale entropy and Zen-NAS are fundamentally different. Zen-NAS uses the gradient norm of the input image as ranking score and proposes to use two feed-forward inferences to approximate the gradient norm for classification. In contrast, ZenDet uses entropy-based score which only requires one feed-forward inference, doubling the search speed.

### 4.2 MULTI-SCALE ENTROPY PRIOR FOR OBJECT DETECTION

In real-world images, the distribution of object size is not uniform. To bring in this prior knowledge, the detection backbone has 5 stages where each stage downsamples the feature resolution to half. The MSEP collects feature maps from the final output of each stage and weighted-sum the corresponding feature map entropies as a new measurement. We name this new measurement as multi-scale entropy. The overall process is illustrated in Figure 1 (b). In this figure, the backbone

extracts multi-scale features $\boldsymbol{C} = (C1, C2, ..., C5)$ at different resolutions. Then the FPN neck fuses $\boldsymbol{C}$ as input features $\boldsymbol{P} = (P1, P2, ..., P7)$ for the detection head. The multi-scale entropy $Z(F)$ of backbone $F$ is then defined by

$$Z(F) := \alpha_1 H(C1) + \alpha_2 H(C2) + \cdots + \alpha_5 H(C5) \tag{5}$$

where $H(Ci)$ is the entropy of $Ci$ for $i = 1, 2, \cdots, 5$. The weights $\boldsymbol{\alpha} = (\alpha_1, \alpha_2, \cdots, \alpha_5)$ store the multi-scale entropy prior to balance the expressivity of different scale features.

**Weights $\alpha$ selection strategy**  As a concrete example in Fig. 1(b), the parts of P3 and P4 are generated by up-sampling of P5, and P6 and P7 are directly generated by down-sampling of P5 (generated by C5). Meanwhile, based on the fact that C5 carries sufficient context for detecting objects on various scales (Chen et al., 2021a), C5 is important in the backbone search, so it is good to set a larger value for the weight $\alpha_5$. Then, different combinations of $\alpha$ and correlation analysis are explored in Appendix D, indicating that $\boldsymbol{\alpha} = (0, 0, 1, 1, 6)$ is good enough for the FPN structure.

### 4.3 ZenDet-NAS in Evolutionary Algorithm Framework

---
**Algorithm 1** ZenDet-NAS with Coarse-to-Fine Evolution
---
**Require:** Search space $\mathcal{S}$, inference budget $B$, maximal depth $L$, total number of iterations $T$, evolutionary population size $N$, initial structure $F_0$, fine-search flag *Flag*.
**Ensure:** NAS-designed ZenDet backbone $F^*$.
 1: Initialize population $\mathcal{P} = \{F_0\}$, *Flag=False*.
 2: **for** $t = 1, 2, \cdots, T$ **do**
 3:   **if** $t$ equals to $T/2$ **then**
 4:     Keep top 10 networks of highest multi-scale entropy in $\mathcal{P}$ and remove the others.
 5:     Set $Flag = True$.
 6:   **end if**
 7:   Randomly select $F_t \in \mathcal{P}$.
 8:   Mutate $\hat{F}_t = \text{MUTATE}(F_t, \mathcal{S}, Flag)$
 9:   **if** $\hat{F}_t$ exceeds inference budget or has more than $L$ layers **then**
10:     Do nothing.
11:   **else**
12:     Get multi-scale entropy $Z(\hat{F}_t)$.
13:     Append $\hat{F}_t$ to $\mathcal{P}$.
14:   **end if**
15:   Remove networks of the smallest multi-scale entropy if the size of $\mathcal{P}$ exceeds $B$.
16: **end for**
17: Return $F^*$, the network of the highest multi-scale entropy in $\mathcal{P}$.
---

---
**Algorithm 2** MUTATE
---
**Require:** Structure $F_t$, search space $\mathcal{S}$, fine-search flag *Flag*.
**Ensure:** Randomly mutated structure $\hat{F}_t$.
 1: Uniformly select a block $h$ in $F_t$.
 2: **if** *Flag* equals to *True* **then**
 3:   Uniformly alternate the kernel size, width within some range.
 4: **else**
 5:   Uniformly alternate the block type, kernel size, width and depth within some range.
 6: **end if**
 7: Return the mutated structure $\hat{F}_t$.
---

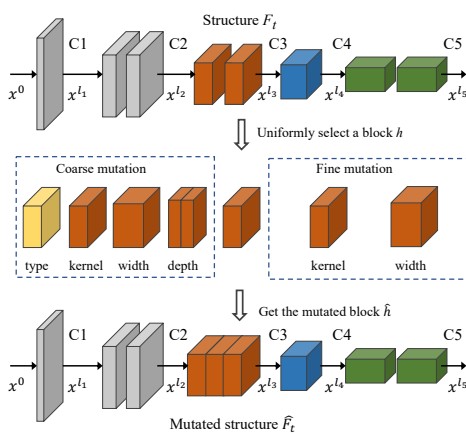

Figure 2: Visualization of Algorithm 2.

Combining all above, we present our ZenDet-NAS in Algorithm 1. The ZenDet-NAS maximizes

the multi-scale differential entropy of detection backbones using Evolutionary Algorithm (EA). To improve the evolution efficiency, a coarse-to-fine strategy is proposed to reduce the search space gradually. First, we randomly generate $N$ seed architectures to fill the population $\mathcal{P}$. As shown in Figure 2, a seed architecture $F_t$ consists of a sequence of building blocks such as ResNet block (He et al., 2016) or MobileNet block (Sandler et al., 2018). Then we randomly select one block and replace it with its mutated version. We use coarse-mutation in the early stage of EA and switch to fine-mutation after $T/2$ EA iterations. In the coarse-mutation, the block type, kernel size, depth and width are mutated randomly; in the fine-mutation, only kernel size and width are mutated.

After the mutation, if the inference cost of the new structure $\hat{F}_t$ does not exceed the budget (e.g., FLOPs, parameters and latency) and its depth is smaller than budget $L$, $\hat{F}_t$ is appended into the population $\mathcal{P}$. The maximal depth $L$ prevents the algorithm from generating over-deep structures. During EA iterations, the population is maintained to a certain size by discarding the worst candidate of the smallest multi-scale entropy. At the end of ZenDet-NAS, the backbone with the highest multi-scale entropy is returned.

## 5 EXPERIMENTS

In this section, we first describe detail settings for searching and training with ZenDet-NAS. Then in subsection 5.2, we apply ZenDet-NAS to design better ResNet-like backbones on COCO dataset (Lin et al., 2014). We align the inference budget with ResNet-50/101. The performance of ZenDet and ResNet are compared under multiple detection frameworks including RetinaNet (Lin et al., 2017b), FCOS (Tian et al., 2019), and GFLV2 (Li et al., 2021). For fairness, we use the same training setting in all experiments for all backbones. In subsection 5.3, we compare the search cost of ZenDet to SOTA NAS methods for object detection. Subsection 5.4 reports the ablation studies of different components in ZenDet-NAS. Finally, subsection 5.5 verifies the transferability of ZenDets on several detection datasets and segmentation tasks.

### 5.1 EXPERIMENT SETTINGS

**Searching Details** In ZenDet-NAS, the evolutionary population $N$ is set to 256. The total EA iterations $T = 96000$. Following the previous designs (Chen et al., 2019b; Jiang et al., 2020; Du et al., 2020), ZenDet is optimized for FLOPs. The resolution for computing entropy is $384 \times 384$.

**Dataset and Training Details** We evaluate detection performance on COCO (Lin et al., 2014) using the official training/testing splits. The mAP is evaluated on val 2017 by default and GFLV2 is additionally evaluated on test-dev 2007 following common practice. All models are trained from scratch (He et al., 2019) for 6X (73 epochs) on COCO. Following the Spinenet (Du et al., 2020), we use multi-scale training and Synchronized Batch Normalization (SyncBN). For VOC dataset, train-val 2007 and train-val 2012 are used for training, and test 2007 for evaluation. For image classification, all models are trained on ImageNet-1k (Deng et al., 2009) with a batch size of 256 for 120 epochs. Other setting details can be found in Appendix A.

### 5.2 DESIGN BETTER RESNET-LIKE BACKBONES

We search efficient ZenDet backbones for object detection and align with ResNet-50/101 in Table 1. ZenDet-S uses 60% less FLOPs than ResNet-50; ZenDet-M is aligned with ResNet-50 with similar FLOPs and number of parameters as ResNet-50; ZenDet-L is aligned with ResNet-101. The feature dimension in the FPN and heads is set to 256 for ZenDet-M and ZenDet-L but is set to 192 for ZenDet-S. The fine-tuned results of models pre-trained on ImageNet-1k are reported in Appendix C.

In Table 1, ZenDet outperforms ResNet by a large margin. The improvements are consistent across three detection frameworks. Particularly, when using the newest framework GFLV2, ZenDet improves COCO mAP by $+2\%$ at the similar FLOPs of ResNet-50, and speeds up the inference by 1.54x times faster at the same accuracy as ResNet-50. Figure 3 visualizes the comparison in Table 1.

**Remark** Please note that we did not copy the numbers of baseline methods reported in previous works in Table 1 because the mAP not only depends on the architecture but also depends on the training schedule, such as training epochs, learning rate, pre-training and so on. Therefore, for a fair

Table 1: ZenDet and ResNet on the COCO. All results using the same training setting. FPS on V100 is benchmarked on the full model with NVIDIA V100 GPU, pytorch, FP32, batch size 32.

| Backbone | FLOPs Backbone | Params Backbone | Head | val2017 $\mathbf{AP_{val}}$ | $\mathbf{AP}_S$ | $\mathbf{AP}_M$ | $\mathbf{AP}_L$ | test-dev $\mathbf{AP_{test}}$ | FPS on V100 |
|---|---|---|---|---|---|---|---|---|---|
| R50 | 83.6G | 23.5M | RetinaNet | 40.2 | 24.3 | 43.3 | 52.2 | - | 23.2 |
| | | | FCOS | 42.7 | 28.8 | 46.2 | 53.8 | - | 27.6 |
| | | | GFLV2 | 44.7 | 29.1 | 48.1 | 56.6 | 45.1 | 24.2 |
| R101 | 159.5G | 42.4M | RetinaNet | 42.1 | 25.8 | 45.7 | 54.1 | - | 18.7 |
| | | | FCOS | 44.4 | 28.3 | 47.9 | 56.9 | - | 21.6 |
| | | | GFLV2 | 46.3 | 29.9 | 50.1 | 58.7 | 46.5 | 19.4 |
| ZenDet-S | 48.7G | 21.2M | RetinaNet | 40.0 | 23.9 | 43.3 | 52.7 | - | 35.5 |
| | | | FCOS | 42.5 | 26.8 | 46.0 | 54.6 | - | 43.0 |
| | | | GFLV2 | 44.7 | 27.6 | 48.4 | 58.2 | 44.8 | 37.2 |
| ZenDet-M | 89.9G | 25.8M | RetinaNet | 42.0 | 26.7 | 45.2 | 55.1 | - | 21.5 |
| | | | FCOS | 44.5 | 28.6 | 48.1 | 56.1 | - | 24.2 |
| | | | GFLV2 | 46.8 | 29.9 | 50.4 | 60.0 | 46.7 | 22.2 |
| ZenDet-L | 152.9G | 43.9M | RetinaNet | 43.0 | 27.3 | 46.5 | 56.0 | - | 17.6 |
| | | | FCOS | 45.9 | 30.2 | 49.4 | 58.4 | - | 19.2 |
| | | | GFLV2 | 47.6 | 30.2 | 51.8 | 60.8 | 48.0 | 18.1 |

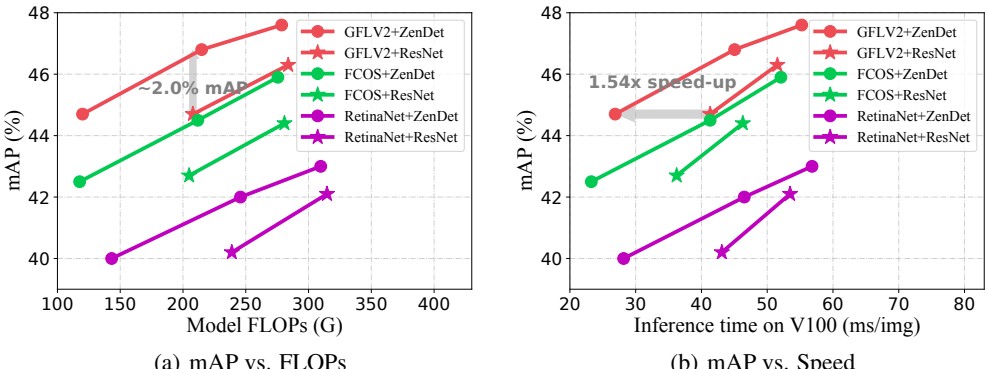

(a) mAP vs. FLOPs      (b) mAP vs. Speed

Figure 3: mAP vs. FLOPs and inference speed on COCO val 2017 in Table 1. Note that FLOPs in (a) is the value of the full detector, cotaining backbone, FPN and head.

comparison, all models in Table 1 are trained by the same training schedule. For comparison with numbers reported in previous works, see subsection 5.3.

## 5.3 COMPARISON WITH SOTA NAS METHODS

Table 2: Comparisons with SOTA NAS methods for object detection. FLOPs are counted for full detector.

| Method | Training-free | Search Cost GPU Days | Search Part | FLOPs All | Pretrain/ Scratch | Epochs | COCO ($\mathbf{AP_{test}}$) |
|---|---|---|---|---|---|---|---|
| DetNAS | × | 68 | backbone | 289G | Pretrain | 24 | 43.4 |
| SP-NAS | × | 26 | backbone | 655G | Pretrain | 24 | 47.4 |
| SpineNet | × | 100x TPUv3† | backbone+FPN | 524G | Scratch | 350 | 48.1 |
| ZenDet | ✓ | **0.6** | backbone | 279G | Scratch | 73 | 48.0 |

†: SpineNet paper did not report the total search cost, only mentioned that 100 TPUv3 was used.

In Table 2, we compare ZenDet with SOTA NAS methods for the backbone design in object detection. We directly use the numbers reported in the original papers. Since each NAS method uses different design spaces and training settings, it is impossible to make an absolutely fair comparison for all methods that everyone agrees with. Nevertheless, we list the total search cost, mAP and

Table 3: Comparisons between ZenDet, DetNAS (Chen et al., 2019b) and SpineNet (Du et al., 2020) under the same training settings. All backbones are trained under GFLV2 head with 6X training epochs. FLOPs and parameters are counted for full detector.

| Backbone | Search Part | Search Space | FLOPs | Params | $\mathbf{AP_{val}}$ | $\mathbf{AP}_S$ | $\mathbf{AP}_M$ | $\mathbf{AP}_L$ | FPS on V100 |
|---|---|---|---|---|---|---|---|---|---|
| DetNAS-3.8G | backbone | ShuffleNetV2 +Xception | 205G | 35.5M | 46.4 | 29.3 | 50.0 | 59.0 | 17.6 |
| SpineNet-96 | backbone+FPN | ResNet Block | 216G | 41.3M | 46.6 | 29.8 | 50.2 | 58.9 | 19.9 |
| ZenDet-M | backbone | ResNet Block | 215G | 34.9M | 46.8 | 29.9 | 50.4 | 60.0 | 22.2 |

FLOPs of the best models reported in each work. This gives us an overall impression of how each NAS method works in real-world practice. From Table 2, ZenDet is the only zero-shot (training-free) method with $48.0\%$ mAP on COCO, using $0.6$ GPU days of search. SpineNet (Du et al., 2020) achieves a slightly better mAP with 2x more FLOPs. It uses 100 TPUv3 for searching but the total search cost is not reported in the original paper. ZenDet achieves better mAP than DetNAS (Chen et al., 2019b) and SP-NAS(Jiang et al., 2020) while being $50 \sim 100$ times faster in search.

To further fairly compare different backbones under the same training settings, we train backbones designed by ZenDet-NAS and previous backbone NAS methods in Table 3. Because the implementation of SP-NAS is not open-sourced, we re-train ZenDet, DetNAS and SpineNet on COCO from scratch. Table 3 shows that ZenDet requires fewer parameters and has a faster inference speed on V100 when achieving the competitive performance over DetNAS and SpineNet on COCO.

## 5.4 ABLATION STUDY AND ANALYSIS

Table 4: Comparison of different evolutionary searching strategies in ZenDet-NAS. C-to-F: Coarse-to-Fine. Zen-Score is the proxy in Zen-NAS (Lin et al., 2021).

| | | ImageNet-1K | | | COCO with YOLOF | | | | COCO with FCOS | | | |
|---|---|---|---|---|---|---|---|---|---|---|---|---|
| Score | Mutation | FLOPs | Params | TOP-1 % | $\mathbf{AP_{val}}$ | $\mathbf{AP}_S$ | $\mathbf{AP}_M$ | $\mathbf{AP}_L$ | $\mathbf{AP_{val}}$ | $\mathbf{AP}_S$ | $\mathbf{AP}_M$ | $\mathbf{AP}_L$ |
| R50 | None | 4.1G | 23.5M | 78.0 | 37.8 | 19.1 | 42.1 | 53.3 | 38.0 | 23.2 | 40.8 | 47.6 |
| Zen-Score | Coarse | 4.4G | 67.9M | 78.9 | 38.9 | 19.0 | 43.2 | 56.0 | 38.1 | 23.2 | 40.5 | 48.1 |
| Single-scale | Coarse | 4.4G | 60.1M | 78.7 | 39.8 | 19.9 | 44.4 | 56.5 | 38.8 | 23.1 | 41.4 | 50.1 |
| Multi-scale | Coarse | 4.3G | 29.4M | 78.9 | 40.1 | 21.1 | 44.5 | 55.9 | 39.4 | 23.7 | 42.3 | 50.0 |
| Multi-scale | C-to-F | 4.4G | 25.8M | **79.1** | **40.3** | **20.8** | **44.7** | **56.4** | **40.0** | **24.5** | **42.6** | **50.6** |

Table 4 reports the ZenDet backbones searched by different evolutionary strategies and whether using multi-scale entropy prior. The COCO mAPs of models trained in two detection frameworks (YOLOF and FCOS) are reported in the right big two columns. YOLOF models are trained by 12 epochs with ImageNet pre-trained initialization, while FCOS models are trained with the 3X training epochs. We also compare their image classification ability on ImageNet-1k. All models are constrained by the FLOPs less than 4.4 G while the number of parameters is not constrained. More details about the searching process and architectures could be found in Appendix G, H.

**Single-scale score** Compared to ResNet-50, the model searched by single-scale entropy score obtains $+0.7\%$ accuracy gain on ImageNet, $+2\%$ mAP gain with FPN-free YOLOF and $+0.8\%$ mAP gain with FPN-based FCOS. Meanwhile, the model searched by Zen-Score achieves $+0.9\%$ accuracy gain on ImageNet, $+1.1\%$ mAP gain with YOLOF and $+0.1\%$ mAP gain with FCOS.

**Multi-scale entropy score** When using multi-scale entropy, both single-scale model and multi-scale model get similar accuracy on ImageNet. The single-scale model uses 2X more parameters than the multi-scale model under the same FLOPs constraint. In terms of mAP, multi-scale model outperforms single-scale model by $+0.3\%$ on COCO with YOLOF and $0.6\%$ on COCO with FCOS. From the last row of Table 4, the coarse-to-fine mutation further enhances the performance of multi-scale entropy prior, and the overall improvement over ResNet-50 is $+1.1\%$ on ImageNet-1k, $+2.5\%$ on COCO with YOLOF and $+2.0\%$ on COCO with FCOS.

**Correlations during the search** To further study the correlations between mAP and scores, models during the search are trained and the results are exhibited in Figure 4. The right part of Figure 4 indicates that the mAP positively correlates with the multi-scale entropy score. The left part

of Figure 4 reveals that the single-scale entropy score cannot represent the mAP well, so multi-scale entropy is necessary for detection tasks. By analysing the structures in Appendix G, the computation of single-scale models is concentrated in the last stage C5, ignoring the C3 and C4 stages, and leading the worse multi-scale score. Instead, multi-scale models allocate more computation to the previous stages to enhance the expressivity of C3 and C4, which improve the multi-scale score.

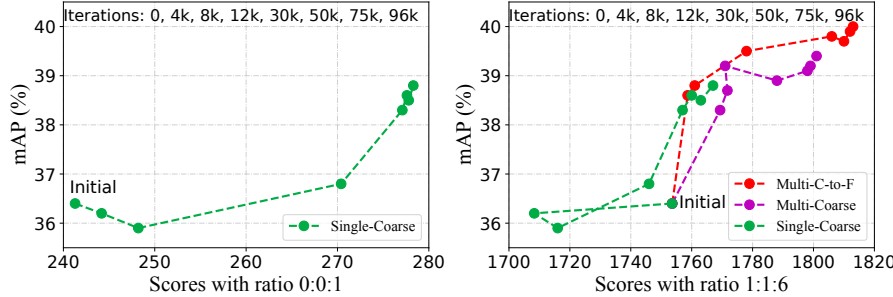

Figure 4: mAP (on FCOS) vs. scores during the search with different search strategies. The scores on the left and the right are computed with the ratio of 0:0:1 and 1:1:6 respectively. **Starting from the initial point, the dotted line indicates the evolution direction in the search process**.

## 5.5 TRANSFER LEARNING

Table 5: Transferability of ZenDet in multiple object detection and instance segmentation tasks. FLOPs reported are counted for full detector.

| Task | Dataset | Head | Backbone | Resolution | Epochs | FLOPs | $AP_{val}$ | $AP_{val}^{mask}$ |
|---|---|---|---|---|---|---|---|---|
| Object Detection | VOC | FCOS | R50 | $1000 \times 600$ | 12 | 120G | 76.8 | - |
| | | | ZenDet-M | $1000 \times 600$ | 12 | 123G | **80.9** | - |
| | Citescapes | | R50 | $2048 \times 1024$ | 64 | 411G | 37.0 | - |
| | | | ZenDet-M | $2048 \times 1024$ | 64 | 426G | **38.1** | - |
| Instance Segmentation | COCO | MASK R-CNN | R50 | $1333 \times 800$ | 73 | 375G | 43.2 | 39.2 |
| | | | ZenDet-M | $1333 \times 800$ | 73 | 379G | **44.5** | **40.3** |
| | | | R50† | $640 \times 640$ | 350 | 228G | 42.7 | 37.8 |
| | | | SpineNet-49† | $640 \times 640$ | 350 | 216G | 42.9 | 38.1 |
| | | SCNet | R50 | $1333 \times 800$ | 73 | 671G | 46.3 | 41.6 |
| | | | ZenDet-M | $1333 \times 800$ | 73 | 675G | **47.1** | **42.3** |

†: Numbers are cited from SpineNet paper (Du et al., 2020).

**VOC and Cityscapes**     To evaluate the transferability of ZenDet in different datasets, we transfer the FCOS-based ZenDet-M to VOC and Cityscapes dataset, as shown in the upper half of Table 5. The models are fine-tuned after pre-trained on ImageNet. Comparing to ResNet-50, ZenDet-M achieves +4.1% better mAP in VOC, +1.1% better mAP in Cityscape.

**Instance Segmentation**     The lower half of Table 5 reports results of Mask R-CNN (He et al., 2017) and SCNet (Vu et al., 2021) models for the COCO instance segmentation task with 6X training from scratch. Comparing to ResNet-50, ZenDet-M achieves better AP and mask AP with similar model size and FLOPs on Mask RCNN and SCNet.

## 6 CONCLUSION

In this paper, we propose a zero-shot NAS method termed ZenDet-NAS for designing high performance backbones for object detection. While achieving better or competitive detection accuracy, the search speed of ZenDet-NAS is several orders of magnitude faster than previous training-based NAS methods. Within one GPU day, the ZenDet automatically designed by ZenDet-NAS is significantly more efficient than popular SOTA backbones for object detection in terms of FLOPs and inference latency. Extensive experiments and analyses on various datasets validate its excellent transferability. In the future, we will generalize ZenDet-NAS to search for detection backbone and FPN neck in a unified framework.

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

## A    TRAINING DETAILS

**Searching Details**    In ZenDet-NAS, the evolutionary population $N$ is set as 256 while total iterations $T = 96000$. Residual blocks and bottleneck blocks are utilized as searching space when comparing with ResNet series backbone (He et al., 2016). Following the previous designs (Chen et al., 2019b; Jiang et al., 2020; Du et al., 2020), ZenDet is optimized for the budget of FLOPs according to the target networks, i.e., ResNet-50 and ResNet-101. To balance the computational complexity and large resolution demand, the resolution in search is set as $384 \times 384$ for ZenDet. When starting the search, the initial structure is composed of 5 downsampling stages with small and narrow blocks to meet the reasoning budget. In the mutation, whether the coarse-mutation or fine-mutation, the width of the selected block is mutated in a given scale $\{1/1.5, 1/1.25, 1, 1.25, 1.5, 2\}$, while the depth increases or decreases 1 or 2. The kernel size is searched in set $\{3, 5\}$. Note that blocks deeper than 10 will be divided into 2 blocks equally to enhance the diversity.

**Dataset and Training Details**    For object detection, trainval35k with 115K images in the COCO dataset is mainly used for training. With the single-scale testing of resolution $1333 \times 800$, COCO mAP results are reported on the val 2017 for most experiments and the test-dev 2007 for GFLV2 results in Table 1. When training on the COCO dataset, the initial learning rate is set to 0.02, and decays two times with the ratio of 0.1 during training. SGD is adopted as optimizer with momentum 0.9; weight decay of $10^{-4}$; batch size of 16 (on 8 Nvidia V100 GPUs); patch size of $1333 \times 800$.

Additionally, multi-scale training and Synchronized Batch Normalization (SyncBN) are adopted to enhance the stability of the scratch training without increasing the complexity of inference. Training from scratch is used to avoid the gap between ImageNet pre-trained model, to ensure a fair comparison with baselines. 3X learning schedule is applied for the ablation study with a multi-scale range between $[0.8, 1.0]$ (36 epochs, decays at 28 and 33 epochs), and 6X learning schedule for the SOTA comparisons with the range between $[0.6, 1.2]$ (73 epochs, decays at 65 and 71 epochs). All object detection training is produced under mmdetection (Chen et al., 2019a) for fair comparisons, and hyperparameters not mentioned in the paper are always set to default values in mmdection.

For image classification, all models are trained on ImageNet-1K with a batch size of 256 for 120 epochs. When training on ImageNet-1K, We use SGD optimizer with momentum 0.9; cosine learning rate decay (Loshchilov & Hutter, 2017); initial learning rate 0.1; weight decay $4 \times 10^{-5}$.

## B    ZENDET-NAS FOR MOBILE DEVICE

Table 6: ZenDet-MB and MobileNetV2 on the COCO with the SSDLite head, which are trained from scratch with 600 epochs at resolution 320. FPS on Pixel 2 is benchmarked on the full model with CPU, FP32, batch size 1. ZenDet-MB-M-SE means inserting SE modules to ZenDet-MB-M.

| Backbone | FLOPs Backbone | Params Backbone | $\mathbf{AP_{val}}$ | $\mathbf{AP}_S$ | $\mathbf{AP}_M$ | $\mathbf{AP}_L$ | FPS on Pixel 2 |
|---|---|---|---|---|---|---|---|
| MobileNetV2-0.5 | 217M | 0.7M | 14.7 | 0.8 | 11.0 | 31.2 | 13.5 |
| MobileNetV2-1.0 | 651M | 2.2M | 21.1 | 1.7 | 20.5 | 39.9 | 6.6 |
| ZenDet-MB-S | 201M | 0.6M | 15.9 | 0.8 | 12.2 | 31.8 | 13.8 |
| ZenDet-MB-M | 645M | 2.0M | 22.2 | 2.1 | 21.5 | 42.3 | 6.3 |
| ZenDet-MB-M-SE | 647M | 2.3M | 22.6 | 2.3 | 22.0 | 42.5 | 5.6 |

For mobile-size object detection, we explore building ZenDet-MB with MobileNetV2 (Sandler et al., 2018) blocks, using the inverted bottleneck block with expansion ratio of 1/3/6. The weight ratio $\alpha$ is still set as 1:1:6 and other searching settings are the same as the Resnet-like searching. In Table. 6, ZenDet-MB use less computation and parameters but outperform MobileNetV2 by 1% AP with similar inference time on Google Pixel 2 phone. Additionally, inserting SE modules to ZenDet-MB-M could improve the mAP by 0.4%.

## C    OBJECT DETECTION WITH IMAGENET PRE-TRAIN MODELS

In the main body of the paper, training from scratch is used to avoid the gap between ImageNet pre-trained model, to ensure a fair comparison with baselines He et al. (2019). Since 6X training

from scratch inevitably consumes 3 times more time than 2X pre-trained training, we use the ImageNet pre-trained model to initialize the ZenDet-M in various heads, including RetinaNet, FCOS and GFLV2. As present in Table. 1, 7, whether using training from scratch or ImageNet pre-training, ZenDet could outperform ResNet-50 in the three popular FPN-based frameworks by large margins.

Table 7: Results between Scratch and Pretrain strategy on the COCO with single-scale testing. Training strategy on ImageNet is same as Table.4.

| Backbone | FLOPs Backbone | Params Backbone | Head | Strategy | Epochs | $AP_{val}$ | $AP_S$ | $AP_M$ | $AP_L$ |
|---|---|---|---|---|---|---|---|---|---|
| R50 | 83.6G | 23.5M | GFLV2 | Scratch | 73 | 44.7 | 29.1 | 48.1 | 56.6 |
| | | | GFLV2 | Pretrain | 24 | 44.0 | 27.1 | 47.8 | 56.1 |
| | | | GFLV2 | Pretrain | 24 | 43.9† | - | - | - |
| ZenDet-M | 89.9G | 25.8M | RetinaNet | Scratch | 73 | 42.0 | 26.7 | 45.2 | 55.1 |
| | | | RetinaNet | Pretrain | 24 | 42.3 | 25.3 | 46.5 | 56.0 |
| | | | FCOS | Scratch | 73 | 44.5 | 28.6 | 48.1 | 56.1 |
| | | | FCOS | Pretrain | 24 | 44.5 | 28.8 | 48.5 | 56.9 |
| | | | GFLV2 | Scratch | 73 | 46.8 | 29.9 | 50.4 | 60.0 |
| | | | GFLV2 | Pretrain | 24 | 46.0 | 29.0 | 50.0 | 59.9 |

†: results in this line are reported in the official github (Li et al., 2021).

## D    WEIGHTS ARRANGEMENT

Table 8: Results between different arrangements of weights in MSEP on COCO. R50 represents ResNet-50.

| Backbone | $\alpha_3:\alpha_4:\alpha_5$ | FLOPs | Params | $AP_{val}$ | $AP_{50}$ | $AP_{75}$ | $AP_S$ | $AP_M$ | $AP_L$ |
|---|---|---|---|---|---|---|---|---|---|
| R50 | None | 83.6G | 23.5M | 38.0 | 55.2 | 41.0 | 23.2 | 40.8 | 47.6 |
| ZenDet | 1:1:1 | 84.4G | 11.5M | 37.4 | 54.6 | 40.0 | 23.6 | 39.8 | 46.6 |
| ZenDet | 1:1:2 | 84.8G | 13.4M | 37.8 | 54.9 | 40.5 | 23.2 | 40.0 | 47.8 |
| ZenDet | 1:1:4 | 85.9G | 17.2M | 38.6 | 56.0 | 41.4 | 23.4 | 41.3 | 48.6 |
| ZenDet | 1:1:6 | 88.7G | 29.4M | **39.4** | **57.3** | **42.1** | **23.7** | 42.3 | **50.0** |
| ZenDet | 1:1:8 | 89.9G | 31.7M | **39.4** | 57.2 | 42.0 | **23.7** | **42.5** | 49.5 |
| ZenDet | 1:4:1 | 86.3G | 10.9M | 35.7 | 52.6 | 38.3 | 22.2 | 38.1 | 44.9 |
| ZenDet | 4:1:1 | 86.1G | 11.1M | 33.9 | 50.2 | 36.7 | 20.4 | 36.1 | 43.4 |

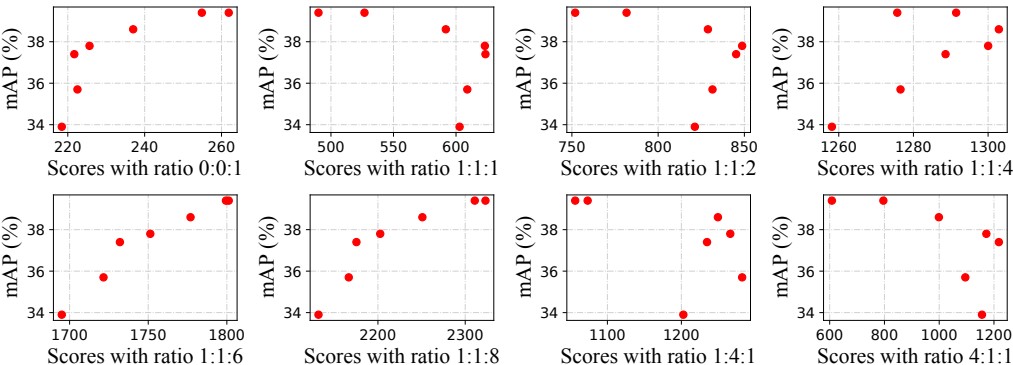

Figure 5: mAP vs. scores. All models are from Table 8 and the scores are computed with different weight ratios. When the ratio is equal to 1:1:6, the correlation between mAP and score is well fitted.

In Table 8, we tune the different arrangements of multi-scale weights in a wide range. Seven multi-scale weight ratios are used to search different models, and all models are trained on the COCO dataset with FCOS and 3X learning schedule. Table 8 shows that if the same weights are arranged to C3-C5, the performance of ZenDet on COCO is worse than ResNet-50. Considering the importance of C5 (discussed in Section 4.2), we increase the weight of C5, and ZenDet's performance continues

to improve. To further explore the correlations between mAP and scores, we use the seven weight ratios to calculate the different scores of each model, along with the single-scale weight ratio of 0:0:1. The correlations between mAP and different scores are plotted in Figure 5. Taking the results in Table 8 and Figure 5, we confirm the ratio of 1:1:6 may be good enough for the current FPN structure.

# E   DISCUSSION ABOUT AUXILIARY COMPONENTS

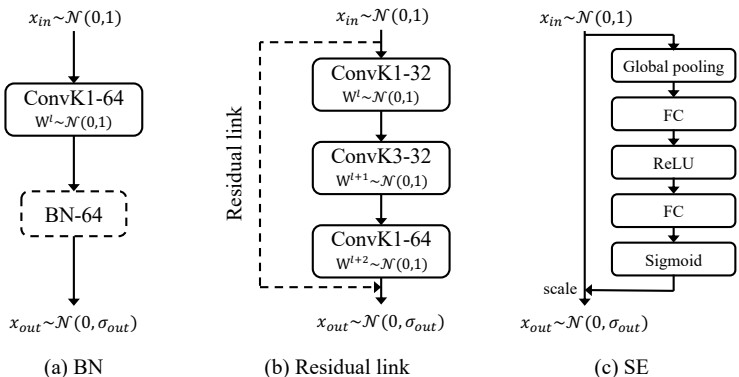

(a) BN                 (b) Residual link                 (c) SE

Figure 6: Build the basic structure of auxiliary components with 64 channel input by standard Gaussian initialization. The resolution of inpput is $192 \times 192$. ConvK1-64 means the convolution with kernel size 1 and output channel 64.

In this section, we explain why auxiliary components are removed during the search and plugged in back for training. Common used three auxiliary components are built in Figure 6 to verify effects on feature variance.

**BN**     BN is a widely used method to re-centre and re-scale the feature to make the network converge faster and more stable. In Figure 6 (a), BN normalize entropy adaptively to the network width (which can be related to output variance). When BN is presented, networks of different widths will have the same entropy value. Hence, BN must be removed when calculating the entropy.

**Residual link**     Building the structure like Figure 6 (b), the variances with or without residual link are 767 or 751 respectively. The residual link brings less than 2% difference in entropy score. The results means that residual link has little effect on entropy, so it could be removed during the search.

**SE**     SE module is used to adaptively recalibrate channel-wise feature responses by explicitly modelling interdependencies between channels Hu et al. (2018). Note that the input is a zero-mean distribution in our computation of the entropy. If SE module is used in the network, the output after global pooling is equal to 0. So, the final output of SE module is equal to 0.5, which loses the ability to model interdependencies between channels.

To sum up, removing these auxiliary components makes our method stable and applicable to most single-branch feed-forward networks.

# F   COMPARISON WITH ZERO-SHOT PROXIES FOR IMAGE CLASSIFICATION

We compare ZenDet with architectures designed by zero-shot proxies for image classification in previous works, including SyncFlow (Tanaka et al., 2020), NASWOT (Mellor et al., 2021), Zen-NAS (Lin et al., 2021). For a fair comparison, All methods use the same search space, FLOPs budget $91\,G$, searching strategy and training schedule. All searched backbones are trained on COCO with the FCOS head and 3X training from scratch. The results are reported in Table. 9.

Among these methods, SyncFlow and NASWOT perform worse than ResNet-50 on COCO albeit they show competitive performance in image classification tasks. Zen-NAS achieves competitive performance over ResNet-50. The ZenDet outperforms Zen-NAS by $+1.3\%$ mAP with slightly fewer FLOPs and nearly one third of parameters.

Table 9: Different zero-shot proxies on COCO with FCOS. All methods use the same search space, FLOPs budget, searching strategy and training schedule.

| Proxy | FLOPs Backbone | Params Backbone | $\mathbf{AP_{val}}$ | $\mathbf{AP}_S$ | $\mathbf{AP}_M$ | $\mathbf{AP}_L$ |
|---|---|---|---|---|---|---|
| R50 | 84G | 23.5M | 38.0 | 23.2 | 40.8 | 47.6 |
| SyncFlow | 90G | 67.4M | 35.6 | 21.8 | 38.1 | 44.8 |
| NASWOT | 88G | 28.1M | 36.7 | 23.1 | 38.8 | 45.9 |
| Zen-NAS | 91G | 67.9M | 38.1 | 23.2 | 40.5 | 48.1 |
| ZenDet | 89G | 25.8M | **39.4** | 23.7 | 42.3 | 50.0 |

## G  VISUALIZATION OF SEARCHING PROCESS

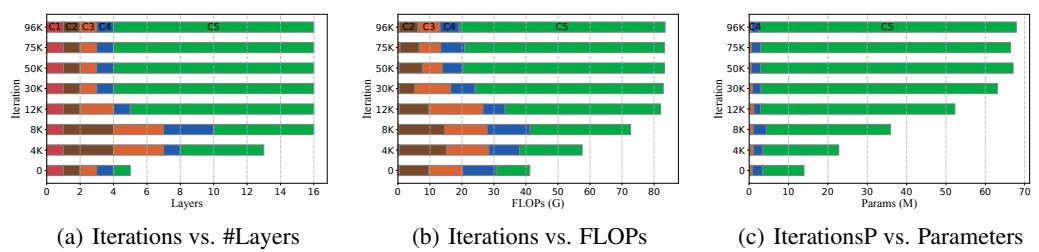

(a) Iterations vs. #Layers  (b) Iterations vs. FLOPs  (c) IterationsP vs. Parameters

Figure 7: Visualization of single-scale entropy searching process. #layer is the number of each block of different levels.

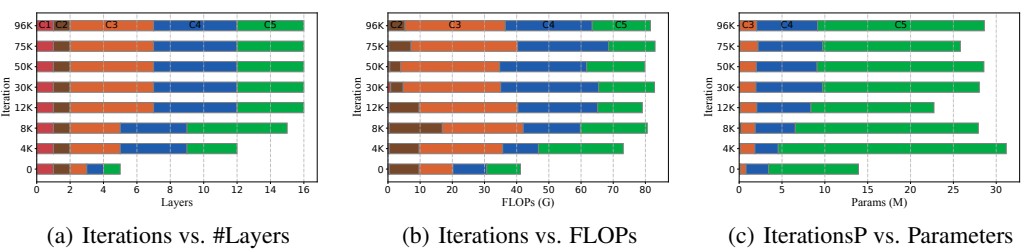

(a) Iterations vs. #Layers  (b) Iterations vs. FLOPs  (c) IterationsP vs. Parameters

Figure 8: Visualization of multi-scale entropy searching process (Coarse). #layer is the number of each block of different levels.

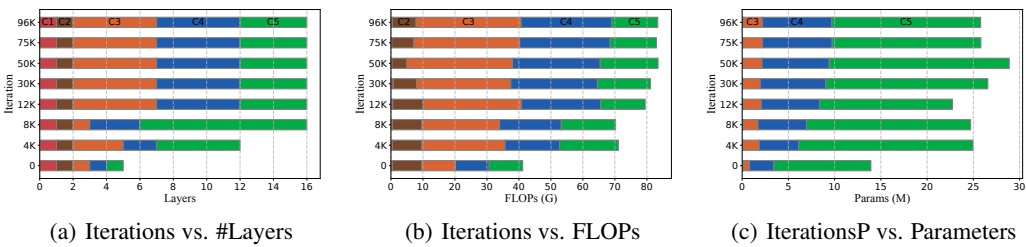

(a) Iterations vs. #Layers  (b) Iterations vs. FLOPs  (c) IterationsP vs. Parameters

Figure 9: Visualization of single-scale entropy searching process (Coarse-to-Fine). #layer is the number of each block of different levels.

The visualizations of searching process on different entropy scores are shown in Figure 7, 8, 9. We compare the change of layers, FLOPs and parameters during different iterations. Visualizations prove our assumptions in the main body of the paper.

# H   DETAIL STRUCTURE OF ZENDETS

We list detail structure in Table 1, 4.

The 'block' column is the block type. 'Conv' is the standard convolution layer followed by BN and RELU. 'ResBlock' is the residual bottleneck block used in ResNet-50 and is stacked by two Blocks in our design. 'kernel' is the kernel size of kxk convolution layer in each block. 'in', 'out' and 'bottleneck' are numbers of input channels, output channels and bottleneck channels respectively. 'stride' is the stride of current block. '# layers' is the number of duplication of current block type.

Table 10: Architecture of single-scale entropy score with coarse mutation in Table. 4

| block | kernel | in | out | stride | bottleneck | # layers | level |
|---|---|---|---|---|---|---|---|
| Conv | 3 | 3 | 96 | 2 | - | 1 | C1 |
| ResBlock | 5 | 96 | 208 | 2 | 32 | 2 | C2 |
| ResBlock | 5 | 208 | 560 | 2 | 56 | 1 | C3 |
| ResBlock | 5 | 560 | 1264 | 2 | 112 | 2 | C4 |
| ResBlock | 5 | 1264 | 1712 | 2 | 224 | 3 | C5 |
| ResBlock | 5 | 1712 | 2048 | 1 | 224 | 3 | C5 |
| ResBlock | 5 | 2048 | 2048 | 1 | 256 | 4 | C5 |

Table 11: Architecture of multi-scale entropy score with coarse mutation in Table. 4

| block | kernel | in | out | stride | bottleneck | # layers | level |
|---|---|---|---|---|---|---|---|
| Conv | 3 | 3 | 32 | 2 | - | 1 | C1 |
| ResBlock | 5 | 32 | 128 | 2 | 24 | 1 | C2 |
| ResBlock | 5 | 128 | 512 | 2 | 72 | 5 | C3 |
| ResBlock | 5 | 512 | 1632 | 2 | 112 | 5 | C4 |
| ResBlock | 5 | 1632 | 2048 | 2 | 216 | 4 | C5 |

Table 12: Architecture of multi-scale entropy score with coarse-to-fine mutation in Table. 4 / ZenDet-M architecture in Table. 1

| block | kernel | in | out | stride | bottleneck | # layers | level |
|---|---|---|---|---|---|---|---|
| Conv | 3 | 3 | 64 | 2 | - | 1 | C1 |
| ResBlock | 3 | 64 | 120 | 2 | 64 | 1 | C2 |
| ResBlock | 5 | 120 | 512 | 2 | 72 | 5 | C3 |
| ResBlock | 5 | 512 | 1632 | 2 | 112 | 5 | C4 |
| ResBlock | 5 | 1632 | 2048 | 2 | 184 | 4 | C5 |

Table 13: ZenDet-S in Table. 1

| block | kernel | in | out | stride | bottleneck | # layers | level |
|---|---|---|---|---|---|---|---|
| Conv | 3 | 3 | 32 | 2 | - | 1 | C1 |
| ResBlock | 5 | 32 | 48 | 2 | 32 | 1 | C2 |
| ResBlock | 3 | 48 | 272 | 2 | 120 | 2 | C3 |
| ResBlock | 5 | 272 | 1024 | 2 | 80 | 5 | C4 |
| ResBlock | 3 | 1024 | 2048 | 2 | 240 | 5 | C5 |

Table 14: ZenDet-L in Table. 1

| block | kernel | in | out | stride | bottleneck | # layers | level |
|---|---|---|---|---|---|---|---|
| Conv | 3 | 3 | 80 | 2 | - | 1 | C1 |
| ResBlock | 3 | 80 | 144 | 2 | 80 | 1 | C2 |
| ResBlock | 5 | 144 | 608 | 2 | 88 | 6 | C3 |
| ResBlock | 5 | 608 | 1912 | 2 | 136 | 6 | C4 |
| ResBlock | 5 | 1912 | 2400 | 2 | 220 | 5 | C5 |

Table 15: Initial structure in the search

| block | kernel | in | out | stride | bottleneck | # layers | level |
|---|---|---|---|---|---|---|---|
| Conv | 3 | 3 | 64 | 2 | - | 1 | C1 |
| ResBlock | 3 | 64 | 256 | 2 | 64 | 1 | C2 |
| ResBlock | 3 | 256 | 512 | 2 | 128 | 1 | C3 |
| ResBlock | 3 | 512 | 1024 | 2 | 256 | 1 | C4 |
| ResBlock | 3 | 1024 | 2048 | 2 | 512 | 1 | C5 |

