# OpenReview forum: "ZenDet: Revisiting Efficient Object Detection Backbones from Zero-Shot Neural Architecture Search"
_ICLR.cc/2022/Conference — ICLR 2022 Submitted_

### Official Review · Reviewer_YY18 · 2021-11-01

**Correctness:** 3
**Technical Novelty And Significance:** 4
**Empirical Novelty And Significance:** 4
**Recommendation:** 6
**Confidence:** 4

**Main Review:**

Strengths
- The proposed method is simple and efficient. This method reduces the search cost largely and speeds up searching.
- This paper is well written and readable.

Weaknesses
- In the experiments, the authors valid the effectiveness of the proposed method on the ResNet Block search space. Could you provide some experiments on other search spaces to validate the generalization of this method?
- During the search stage, residual link, BN and SE are all removed. To my knowledge, these components have some effects on feature variance. Thus, the entropy score based on feature variance can not reflect the performance of networks accurately when removing these components. Further, the network searched by this method may be suboptimal. Could you explain why removing these components?
- In this paper, the smallest backbones searched by this method is ZenDet-S. Due to the limit of edge devices, the detectors with ZenDet-S can not run fast. This raise a question: can this method search some networks for edge devices?
- From Table 5, comparing the row1 and row2, the metric with single-scale score does not outperform the metric with multi-scale score largely. This demonstrates the proposed method does not deeply entangle with detection task. Thus, could you provide an experiment on NAS-Bench-201?
- In Figure 2, there are little details about the search space. Could you provide more details about the search space?

**Summary Of The Paper:**

In this paper, the authors propose a zero-shot NAS to search backbone for detection task. Specifically, this method uses the differential entropy of output features as a metric to measure the performance of architecture on detection task. With the differential entropy , this method does not need training network parameters, reducing the search cost largely. Also, the backbones searched by this method have the state-of-the-art performance on detection task.

**Summary Of The Review:**

I appreciate the contribution of this paper, but I prefer to reject this paper for the above weaknesses. If the authors solve my concerns, I would like to raise my rate.

==Final Decision==\
The author's feedback has resolved my concerns except the ranking. Though the author did not resolve ranking concern, I decide to raise score to 6. I hope the authors would provide the results about how the multi-scale weight ratio affect the correlation for more heads in the final paper.

---

> ### Author Response · Authors · 2021-11-20
> **Responses to Reviewer YY18**
>
> Thank you very much for your encouraging comments. The detailed replies to your comments are listed as follows.
>
> ***
> `Q1:  More search space on more devices. Can this method search some networks for edge devices?`
> `A1:`
> In Appendix B, we build ZenDet with MobileNetV2 blocks for object detection on a mobile device. ZenDet-MB uses less computation and parameters and outperforms MobileNetV2 by 1% AP, with similar inference speed on a Google Pixel2 phone.
> The smallest backbone is ZenDet-MB-S in Appendix B, which could obtain 13.8 FPS on Gooogle Pixel 2 with batch size 1, FP32.
>
> ***
> `Q2: Could you explain why removing these components (residual link, BN and SE)?`
> `A2:`
> * It would be easier to understand our method from the following equivalent process: i) We first search a backbone with only Convolutions; ii) We then add BN layers and residual links, as well as SE block, in the searched backbone for training.
>
> * When computing entropy numerically, any existing BN layers must be removed. This is because BN normalizes entropy adaptively to the network width (which can be related to output variance). When BN is presented, networks of different widths will have the same entropy value.
>
> * The residual link brings less than a 2% difference in entropy score. So removing the residual link does not affect too much on the entropy score.
>
> * The output of SE module is equal to 0.5 due to the random initialization. So SE does not essentially affect the entropy score. In Table.6, inserting SE modules to ZenDet-MB-M could improve the mAP by 0.4%.
>
> Detailed explanations are present in Appendix E.
>
> ***
> `Q3: This demonstrates the proposed method does not deeply entangle with detection tasks. `
> `A3:`
> Since detection is more complex than classification, and Actually, nearly all detection models can be down-graded to image classification models (by simply ignoring the bounding box output). So it is not surprising that ZenDet can be applied to classification models.
>
> However, classification zero-shot methods do not work optimally in detection. To see this, we revise Table.4 In Section 5.4. It is clear that proxies designed for classification may not outperform ZenDet in detection tasks.
>
> ***
> `Q4: Could you provide an experiment on NAS-Bench-201?`
> `A4:`
> NAS-Bench-201 is an excellent work to provide a relatively fair benchmark for the comparison of different NAS algorithms for classification, which can help researchers can focus on designing NAS algorithms while avoiding tedious hyper-parameter tuning.
>
> However, networks in it consist of complex connections and parallel structures. As we have discussed in the above Q&A for Reviewer 34Uv, ZenDet is not yet ready to be directly applied in such design space. In the future, when we solve it, we would like to provide an experiment on this benchmark.
>
> ***
> `Q5:  Could you provide more details about the search space?`
> `A5:` Details about the ResNet-like search space are present in Appendix A. In addition, we add the initial structure in Table 15.
> ***

---

> > ### Comment · Reviewer_YY18 · 2021-11-23
> > **Reviewer Response to Author Response**
> >
> > I have read all the reviews and author's feedback. Based on the rebuttal, the authors have resolved most of my concerns. But I think the authors do not solve my concern in Q2. Though the authors explain why removing the components, like BN and residuals, this removing maybe affect the searched model. In specific, the searched model maybe sub-optimal in performance. To solve this concern, the authors should provide an experiment about rank preserving as suggested by reviewer KnoQ.

---

> > > ### Author Response · Authors · 2021-11-23
> > > **Response to Reviewer YY18**
> > >
> > > Rank preserving is a useful supplementary experiment but technically very difficult, if not even impossible, due to the lack of Detection NAS Database. Please also refer to helpful discussions in Q4&A4 for Reviewer KNoQ.
> > >
> > > To our best efforts, we still provide two similar doable experiments in Figure 4 and Figure 5, in which correlations between mAP and scores are analyzed. The results show that our ZenDet score positively correlates to the detection mAP metric.
> > >
> > > Though removing the auxiliary components may be sub-optimal, this does not change our key conclusion that ZenDet achieves SOTA performance within given computational budgets and is competitive to those training-based NAS methods.

---

> > > > ### Comment · Reviewer_YY18 · 2021-11-29
> > > > **Reviewer Response to Author Response**
> > > >
> > > > As shown in Figure 5, the correlation(ranking) between mAP and scores is affected by the multi-scale weight ratio. In some weight ratios, the correlation is poor. When the ratio is equal to 1:1:6, the correlation is well fitted. But this experiment is only validated with head FCOS. For other heads, like Mask R-CNN, how the multi-scale weight raito affect the correlation?

---

> > > > > ### Author Response · Authors · 2021-11-30
> > > > > **Response to reviewer YY18**
> > > > >
> > > > > Thanks again for your latest response. Since the discussion period is coming to the end, there is not enough time to do this experiment. We will follow your suggestion to create more results with different heads in the future.

---

### Official Review · Reviewer_KNoQ · 2021-11-01

**Correctness:** 3
**Technical Novelty And Significance:** 2
**Empirical Novelty And Significance:** 3
**Recommendation:** 6
**Confidence:** 4

**Main Review:**

Here are some strengths and weaknesses for this submission:

**Strengths:**

+ This paper is clearly written and easy to follow.
+ The proposed method should be fairly easy to implement, and it is appreciated that the authors promise to open source their implementation if the paper is eventually accepted.
+ The paper focuses on a very important aspect in real-world deployment of NAS algorithms: the search cost, and provides an effective solution.
+ The experiments are done mainly on large-scale datasets.

**Weaknesses:**

I. Methodology

+ The paper seems to be largely based on Zen-NAS. I feel that only Section 4.2 of this paper is novel and it seems that the effectiveness is not well justified:
    - How do you choose the weights $\alpha$?
    - It seems that such choice impact the final performance a lot. A bad set of $\alpha$ cannot even outperform manually designed R50. What is the theoretical implication / insight of the current parameter choice?
+ In general the paper mainly justifies their results on accuracy-based ablation studies and comparisons with previous methods. But for me I'm more interested the effectiveness of multi-scale Zen-Score itself.
    - I did not see any ablation analysis studying the correlation between Zen-Score and accuracy of different networks within the design space.
    - I'm also curious whether multi-scale Zen-Score can improve the correlation comparing with single-scale Zen-Score.
    - It will be great if there are some comparisons between ZenDet and one-shot NAS methods in terms of rank preserving.
+ It is well-known that searching the architecture of FPN can also potentially improve the efficienty-accuracy tradeoff of object detectors. I'm curious whether the Zen-Det approach can also be utilized to search for complex connections in the FPN architecture.
+ It'll also be great if the authors can have a small discussion on whether Zen-Det can be applied to recent transformer-based detectors.
+ Removing auxiliary layers such as SE during search and adding them back during training/inference seem to be ad-hoc and not well justified.

II. Experiments

+ The paper is focused on the ResNet design space, which is shown to be efficient on desktop GPUs. However, it seems that results on edge GPUs or CPUs are missing. This makes the experiment part of this paper rather weak comparing with existing work (ZenNAS).
+ Some comparisons are not quite fair / clear in the experiment section:
    - Table 4 claims that the Zen-NAS performance is 38.1 mAP, but in Table 5 it seems that the single-scale version of ZenDet (which I think is exactly Zen-NAS) has 38.8 mAP. This makes me a little bit confused.
    - The comparison between DetNAS and Zen-Det does not seem to be completely fair. The design space is different. Would it be possible to supplement some results on their design space and justify the effectiveness of ZenDet?
+ The optimization target for this paper is FLOPs, but I think targeting latency will be better (and I don't think the change of optimization target will significantly influence the current algorithm pipeline).
    - A natural question related to latency as the optimization target: how well will simpler zero-shot NAS metric work? For example, given a latency budget, if we just randomly select networks whose FLOPs are higher than one certain threshold, will the result be competitive comparing with ZenDet? It will be interesting to present the results on both desktop GPUs and edge devices.

**Summary Of The Paper:**

This paper proposes a zero-shot neural architecture search approach for backbone design in object detection. The idea is to compute an entropy-based multi-scale Zen-score and use the score as an objective for evolutionary architecture search. The paper achieves better results comparing with previous zero-shot NAS approaches on object detection and greatly reduces the search time of conventional NAS approaches while maintaining similar accuracy.

**Summary Of The Review:**

Overall I think this paper presents promising preliminary results on zero-shot neural architecture search for object detection. However, the novelty seems to be limited (comparing with Zen-NAS) and the experiments don't seem to be thorough enough. These downsides prevent me from recommending acceptance at the current stage. However, I'll be glad to raise my score if my concerns are addressed properly during the author response period.

====

Post-rebuttal update: I decided to raise the score to 6 based on the empirical results presented in this paper. I still think the novelty comparing with Zen-NAS is not that large and the challenges of the current approach (e.g. cannot deal with complex connections in the FPN, basically can only search for relatively simple backbones) prevents me from assigning an even higher score to this paper.

---

> ### Author Response · Authors · 2021-11-20
> **Responses to Reviewer KNoQ--Part B**
>
> ***
> `Q7: Removing auxiliary modules during the search seems to be ad-hoc and not well justified.`
> `A7:`
> * It would be easier to understand our method from the following equivalent process: i) We first search a backbone with only Convolutions; ii) We then add BN layers and residual links, as well as SE block, in the searched backbone for training.
>
> * When computing entropy numerically, any existing BN layers must be removed. This is because BN normalizes entropy adaptively to the network width (which can be related to output variance). When BN is presented, networks of different widths will have the same entropy value.
>
> * The residual link brings less than a 2% difference in entropy score. So removing the residual link does not affect too much on the entropy score.
>
> * The output of SE module is equal to 0.5 due to the random initialization. So SE does not essentially affect the entropy score. In Table.6, inserting SE modules to ZenDet-MB-M directly could improve the mAP by 0.4%.
>
> Removing these auxiliary components makes our method stable and applicable to most single-branch feed-forward networks.
>
> Detailed explanations are present in Appendix E.
> ***
> `Q8:  More search space on more devices.`
> `A8:` In Appendix B, we build ZenDet with MobileNetV2 blocks for object detection on a mobile device. ZenDet-MB uses less computation and parameters and outperforms MobileNetV2 by 1% AP, with similar inference speed on a Google Pixel2 phone.
>
> ***
> `Q9: Confusion about ZenDet and Zen-NAS.`
> `A9:` Differences between ZenDet and Zen-NAS are further explained in detail In Section 4.1 (refer to Q1). In Section 5.4, we conduct the comparison between Zen-NAS score and ZenDet on both classification and detection. Results in Table.4 show that the accuracies on ImageNet-1K of the two scores are competitive, but ZenDet obtains better performance than Zen-NAS on detection tasks.
>
> ***
> `Q10: Confusion about the fair comparison between DetNAS and Zen-Det.`
> `A10:`
> * Because the design space is different, the comparison between DetNAS and Zen-Det is not completely fair. To avoid confusion, the caption of Table 3 is revised as "Comparisons between ZenDet, DetNAS and SpineNet under the same training settings".
>
> * We tried our best to make the comparison between DetNAS and ZenDet as fair as possible. To this end, we optimize the training settings for all methods. The reproduced result of DetNAS in Table 3 is slightly better than the original results reported by the DetNAS paper.
>
> * The design space of DetNAS is based on Xception which consists of complex connections and parallel structures. As we have discussed in the above Q&A for Reviewer 34Uv, ZenDet is not yet ready to be directly applied in such design space.
>
> ***
> `Q11: Targeting latency will be better. Randomly select networks whose FLOPs are higher than one certain threshold.`
> `A11:` Following this comment, we constrain the inference speed on GPU V100 and randomly select networks whose FLOPs are higher than 85G. These models are trained on GFLV2 with 3X training schedule. Results will be released later.
> ***

---

> > ### Author Response · Authors · 2021-11-21
> > **Responses to Reviewer KNoQ--Q11 Results**
> >
> > ***
> > `Q11: Targeting latency will be better. Randomly select networks whose FLOPs are higher than one certain threshold.`
> > `A11:` Following this comment, we constrain the inference speed on GPU V100 and randomly select networks whose FLOPs are higher than 85G. These models are trained on GFLV2 with **3X training schedule**, abbreviated as Lat-FLOPs.
> >
> > ***
> > * ResNet-50，   FPS=24.1, FLOPs= 83.6G, Params=23.5M mAP= 41.6
> > * ZenDet-M，    FPS=22.2, FLOPs= 89.9G, Params=25.8M mAP= 43.3
> > ***
> > * Lat-FLOPs_1，FPS=23.6, FLOPs= 97.5G, Params=29.1M mAP= 41.5
> > * Lat-FLOPs_2，FPS=23.5, FLOPs=101.8G, Params=24.6M, mAP= 40.7
> > * Lat-FLOPs_3，FPS=24.4, FLOPs=100.3G, Params=27.1M, mAP= 41.3
> > * Lat-FLOPs_4，FPS=24.3, FLOPs= 99.2G, Params=26.3M, mAP= 41.2
> > ***
> >
> > It is clear that FLOPs is not a good metric in detection. Models with larger FLOPs might even perform worse than ResNet-50.

---

> ### Author Response · Authors · 2021-11-20
> **Responses to Reviewer KNoQ--Part A**
>
> Thank you very much for your encouraging comments. The detailed replies to your comments are listed as follows.
>
> ***
> `Q1: The paper seems to be largely based on Zen-NAS.`
> `A1:` We follow the definition of the vanilla network in Zen-NAS. The rescaling trick to avoid numerical overflow is inspired by Zen-NAS BN layer re-scaling. Otherwise, the two methods overlap little. Most importantly, the principles behind ZenDet and Zen-NAS are fundamentally different.
> * Zen-NAS  is a **gradient-based method for classification** while our ZenDet is an **entropy-based method for detection**.
> * The computational cost for Zen-score used by Zen-NAS is almost **twice** of ZenDet's entropy score.
> * In ablation study, ZenDet works better than Zen-NAS in detection tasks, even using single-scale entropy.
> More details can be found in Sections 4.1 and 5.4.
>
> ***
> `Q2:  The insight of the selection for weight ratio alpha  in multi-scale entropy.`
> `A2:` The values of weight ratio alpha are manually designed, based on the following intuition:
> * The intuitive explanation is related to the structure of FPN. **C5 is more important for the backbone**. Hence, it is good to set a larger value for the weight C5. (More Details can be found in Section 4.2)
> * **Different combinations of alpha and correlations analysis** are explored in Appendix D,  indicating that alpha=(1,1,6) is good enough for the FPN structure.
> * alpha=(1,1,6) works well in all of our experiments, including the ResNet and MobileNetV2 search spaces.
>
> ***
> `Q3: The correlations between the accuracy and entropy scores.`
> A3: To further study the correlations between mAP and scores, models during the search are trained and the results are exhibited in Figure 4.
> * **The mAP positively correlates with the multi-scale entropy score**.
> * The single-scale entropy score cannot represent the mAP well, which shows that multi-scale entropy is necessary for detection tasks.
> Details can be found in Section 5.4 and Appendix G/H.
>
> ***
> `Q4: Comparisons between ZenDet and one-shot NAS methods in terms of rank preserving. `
> `A4:`
> * Regarding rank preserving, it is difficult to evaluate rank preserving in detection tasks because i) there are no existing NAS benchmark datasets for detection tasks; ii) the training cost of the detection model is considerably more expensive.
> * Compared to one-shot NAS methods, the advantage of ZenDet is its fast search speed. Considering the rank preserving, we think one-shot NAS methods may be better because they can finetune sub-models during the search.
>
> ***
> `Q5: Complex connections in the FPN architecture.`
> `A5:` In theory, the Principle of Maximum Entropy is applicable to networks with complex inner layer connections. However, how to ensemble entropies from different layers must be carefully defined, as their numerical scales vary in a very large range. Direct summation may not lead to a meaningful definition of model entropy. On the other hand, the numerical overflow/rescaling problem must be carefully handled in densely connected networks. Properly addressing these new issues in densely connected networks require lots of additional work. Therefore, we leave this open question to future works.
>
> ***
> `Q6: Whether Zen-Det can be applied to recent transformer-based detectors.`
> `A6:`
> * Theoretically, the Principle of Maximum Entropy is applicable to transformers. Currently, the design of transformer is still developing rapidly, and there are no unified vision transformer blocks so far.
> * Training detection transformers is very tricky. Especially, when there are mixtures of transformer blocks and convolutional blocks, current optimizers such as SGD and AdamW are not very stable.
>
> Due to the above difficulties, we leave the zero-shot transformer NAS to future works.
>
> ***

---

> > ### Comment · Reviewer_KNoQ · 2021-11-22
> > **Reviewer Response to Author Response**
> >
> > Thanks for the great efforts in the rebuttal. I'll have to decide whether to increase my score after discussing with other reviewers, but here are my preliminary comments on the author response.
> >
> > A1. Thanks for the clarification. I feel that the term "gradient-based" is a little bit confusing, since it generally refers to gradient-based search in this field.
> >
> > A2-A3. This solves my concern.
> >
> > A4. I don't really think one-shot NAS approaches are strong in terms of rank preserving. I agree that experiments on rank preserving are hard to design, but I think your results in Figure 5 of appendix should be a good example. In order to compare single-scale score with multi-scale score, I believe you only need to experiment with sth. like 0:0:1, and randomly select tens of models, train them from scratch and compare the accuracy. Note that there is no need to supplement new results.
> >
> > A5-A6. Thanks for the clarification. I'm now aware of the challenges for ZenDet.
> >
> > A7. I believe Appendix E is clearly written.
> >
> > A8-A9. Results are impressive.
> >
> > A10. Thanks for the efforts in making a fair comparison between previous methods and yours.
> >
> > A11. I agree with your results, but actually I'm more interested in very small models since previous study [1] indicates that FLOPs is a good performance proxy for tiny models that are designed for microcontrollers. However, there's no need to further supplement new results.
> >
> > [1] Lin et al., MCUNet: Tiny Deep Learning on IoT Devices. In NeurIPS 2020.

---

> > > ### Author Response · Authors · 2021-11-23
> > > **Responses to Reviewer KNoQ**
> > >
> > > Thank you again for your kind response.
> > >
> > > ***
> > > `A1: `Thank you for pointing this out. We will avoid the terminology "gradient-based" and rephrase as follows: " Zen-NAS uses the gradient norm of the input image as ranking score and proposes to use an average of multiple feed-forward inference to approximate the gradient norm. In contrast, ZenDet uses entropy-based score which only requires one feed-forward inference."
> > > ***
> > > `A4: ` We are appreciated to your further idea about the design of rank preserving and will verify it in enough time.
> > > ***
> > > `A11:` We agree that the small FLOPs model is also an important application scenario, and have cited MCUNet in the manuscript. We will try to append this experiment in the final version, using the small-size design space for micro-controllers.
> > > ***

---

### Official Review · Reviewer_34Uv · 2021-11-04

**Correctness:** 3
**Technical Novelty And Significance:** 3
**Empirical Novelty And Significance:** 3
**Recommendation:** 6
**Confidence:** 4

**Main Review:**

Strengths:
1. Saving lots of time and memory for searching architecture for a competitive backbone.
2. Archiving SOTA performance in effective benchmark datasets, especially good performance on COCO 2017 dataset.
3. Benefit from Multi-Scale Entropy Prior, ZenDet can adapt different sizes of objects, and results of training from scratch.
4. In segmentation tasks, ZenDet also outperforms well.

Weakness:
1. No experiments for comparision of SOTA method in Instance Segmentation and results of training from scratch.
2. Existing some spelling mistakes，such as retianet->retinanet.


What are the failure modes of the proposed msep method?

**Summary Of The Paper:**

Detection backbones usually cost a lot. With a novel zero-shot NAS method proposed in this work, named ZenDet, detection tasks can free from the heavy models, time, and resources to archive SOTA performance.

**Summary Of The Review:**

see main review

---

> ### Author Response · Authors · 2021-11-20
> **Responses to Reviewer 34Uv**
>
> Thank you very much for your encouraging comments. The detailed replies to your comments are listed as follows.
>
> ***
> `Q1: No experiments for comparison of SOTA method in Instance Segmentation and results of training from scratch.`
> `A1:`
> In Tab.5, the experiments using MaskRCNN framework for Instance Segmentation are conducted with 6X training from scratch, which is added in this version to avoid confusion.
>
> We have further compared the performance of ResNet-50 and ZenDet-M using the SOTA SCNet (AAAI 2021) framework for Instance Segmentation. Compared to ResNet-50, ZenDet-M achieves better AP and mask AP at similar model sizes and FLOPs when using SCNet. Details can be found in Section 5.5.
>
>
> ***
> `Q2: Existing some spelling mistakes，such as retianet->retinanet.`
> `A2:` Thank you for your careful proofreading. We have thoroughly gone through our paper again to correct typos.
>
> ***
> `Q3: What are the failure modes of the proposed ZenDet method?`
> `A3:` When dealing with complex connections and parallel structures, such as Inception and FPN,  ZenDet is not yet ready to be directly applied in such design space. The challenge is how to ensemble entropies from different layers must be carefully defined, as their numerical scales vary in a very large range. Direct summation may not lead to a meaningful definition of model entropy. Hence, the current ZenDet method is suitable for single-branch feed-forward networks like ResNet or MobileNet which could already satisfy most requirements at present.
> ***

---

### Author Response · Authors · 2021-11-20
**Revision Summary**

We would like to thank anonymous reviewers for their constructive comments and efforts to help improve our work. Considering all the comments, we have thoroughly revised the manuscript. The changes have been highlighted in red font in the revised paper. A summary of the changes is listed as follows.
* In Section 4.1, the fundamental differences between ZenDet and Zen-NAS are explained in detail.
* In Section 4.2, the importance of stage C5 and the selection of the weights alpha are explained along with Appendix D.
* In Section 5.4, we conduct the comparison between Zen-NAS score and ZenDet in classification and detection tasks. The correlations between detection performance and the proposed multi-scale entropy score during the search are explored in detail.
* The subsection "Comparison of Zero-Shot Proxies for Image Classification" is moved to Appendix F (due to space limitation).
* In Section 5.5, we have further compared the performance of R50 with ZenDet-M using one of the SOTA frameworks SCNet (AAAI 2021) for Instance Segmentation.
* In Appendix B, we build ZenDet with MobileNetV2 blocks for mobile-device object detection and measure the inference time on a Google Pixel2 phone.
* In Appendix D, the correlations between performance and different alpha values are visualized to explain the insight of the alpha selection.
* In Appendix E, we explain why and how auxiliary components can be removed during the search.
* We have thoroughly gone through our paper again to correct typos and modify the layout.

Detailed Q&A's are listed below. We look forward to further discussions and feedbacks.

---

### Public Comment · ~Xiuyu_Sun1 · 2022-06-06
**a re-organized paper and its source code**

Thanks to everyone's constructive comments. A re-organized paper 'MAE-DET: Revisiting Maximum Entropy Principle in Zero-Shot NAS for Efficient Object Detection' is now accepted by ICML 2022,  and the arxiv one can be found here https://arxiv.org/abs/2111.13336 . We have also released our source code at https://github.com/alibaba/lightweight-neural-architecture-search

---

### Decision · Program_Chairs · 2022-01-20

**Decision:**

Reject

**Comment:**

This paper received scores of 6,6,6 after the reviewers succeeded in making two authors raise their scores from 5 to 6. However, even after this, none of the reviewers actively argued for the paper. The only positive point raised in the private discussion was that the results are strong. (However, there is still the question of how much of this was due to the different design space used.)
Negative points raised in the private discussion included that
- despite the authors clarification on the differences to Zen-NAS, the difference is perceived not to be large.
- there is no theoretical foundation behind the selection of a critical parameter, and this directly limits the applicability of ZenDet in searching for FPN connections.
- as a paper focused on detection NAS, a limitation to only search for the backbone may not be novel enough for publication at ICLR.

Overall, I agree with this criticism and weakly recommend rejection.